# *Streptococcus pyogenes*: Pathogenesis and the Current Status of Vaccines

**DOI:** 10.3390/vaccines11091510

**Published:** 2023-09-21

**Authors:** Jiachao Wang, Cuiqing Ma, Miao Li, Xue Gao, Hao Wu, Wenbin Dong, Lin Wei

**Affiliations:** Key Laboratory of Immune Mechanism and Intervention on Serious Disease in Hebei Province, Department of Immunology, Hebei Medical University, Shijiazhuang 050017, China

**Keywords:** group A streptococcus, vaccine, rheumatic fever, M protein, antigens, adjuvant

## Abstract

*Streptococcus pyogenes* (group A *Streptococcus*; GAS), a Gram-positive coccal bacterium, poses a significant global disease burden, especially in low- and middle-income countries. Its manifestations can range from pharyngitis and skin infection to severe and aggressive diseases, such as necrotizing fasciitis and streptococcal toxic shock syndrome. At present, although GAS is still sensitive to penicillin, there are cases of treatment failure for GAS pharyngitis, and antibiotic therapy does not universally prevent subsequent disease. In addition to strengthening global molecular epidemiological surveillance and monitoring of antibiotic resistance, developing a safe and effective licensed vaccine against GAS would be the most effective way to broadly address GAS-related diseases. Over the past decades, the development of GAS vaccines has been stalled, mainly because of the wide genetic heterogeneity of GAS and the diverse autoimmune responses to GAS. With outbreaks of scarlet fever in various countries in recent years, accelerating the development of a safe and effective vaccine remains a high priority. When developing a GAS vaccine, many factors need to be considered, including the selection of antigen epitopes, avoidance of self-response, and vaccine coverage. Given the challenges in GAS vaccine development, this review describes the important virulence factors that induce disease by GAS infection and how this has influenced the progression of vaccine development efforts, focusing on several candidate vaccines that are further along in development.

## 1. Introduction

*Streptococcus pyogenes* (GAS; also known as group A Streptococcus) is a Gram-positive coccus that causes a variety of infectious diseases, including (1) superficial infections (purulent tonsillitis, erysipelas, pharyngitis, and cellulitis), (2) toxin-mediated diseases (scarlet fever and streptococcal toxic shock syndrome (STSS)), (3) immune-mediated diseases (acute rheumatic fever (ARF) and rheumatic heart disease (RHD)), and invasive infections (necrotizing fasciitis, bacteremia, and meningitis) [1,2,3]. GAS remains one of the top ten causes of death caused by infections, resulting in a significant global health burden, especially in low- and middle-income countries. The main causes of human death brought about by GAS are autoimmune sequelae (e.g., ARF and RHD) and serious invasive diseases [4,5]. GAS causes damage to host cells, tissues, and the immune system by secreting a large number of virulence factors. For example, the M protein of GAS binds to fibronectin (Fn) on the surface of the host cell to enter epithelial or endothelial cells, which is considered key to the evolutionary success and adaptability of GAS [6]. After colonization, GAS must evade the host’s innate immune system to invade deeper tissue sites and trigger severe infections. Autophagy is one such innate immune defense mechanism against intracellular GAS; however, streptococcal cysteine protease (SpeB) can degrade autophagy adaptor proteins p62, NDP52, and NBR1 to evade the clearance mechanism of the autophagy pathway [7].

At the end of 2022, the World Health Organization (WHO) reported a significant increase in scarlet fever and invasive infections in multiple developed countries, disproportionately affecting children [8,9]. Studies in the United States have shown that between 2005 and 2012, there were an estimated 1136–1607 deaths each year due to GAS infections, and in 2019, there were over 2250 deaths [10]. Furthermore, Canada’s GAS infection rate in 2017 was more than tenfold that of 2003 [11]. Therefore, it is evident that diseases associated with GAS infections are rapidly increasing. At present, penicillins and other beta-lactams are still effective in the treatment of GAS. However, the increasing resistance to other antibiotics used in disease treatment is a growing concern worldwide.

In theory, diseases caused by GAS infection can be effectively prevented via vaccination. However, after decades of research, there is still no licensed and available vaccine. At the 71st World Health Assembly in 2018, strategies for the development of GAS vaccines were recommended to be prioritized [12]. Although new vaccine candidates are entering the development pipeline, they are far from ready for clinical application. This review describes the mechanism of disease induced by the virulence factors of GAS and the current status of its vaccine development, with a special focus on several potential candidate vaccines that are further along in development and have showed the most promise in preclinical studies. We also highlight many challenges and potential development trends for the future.

## 2. GAS Virulence Factors and Pathogenesis

The genome sequences of GAS encode multiple virulence factors, which are crucial for tissue cell colonization and invasive disease progression [2,13]. Understanding the various virulence mechanisms of GAS will help us better understand the causes of disease progression and improve vaccine design and development.

### 2.1. M Protein

The M protein is an α-helical dimer located on the GAS surface. The basic structural components of the M protein consist of an N-terminal hypervariable region, central domain, and C-terminal conserved region [14]. Due to the high variability of the N terminus of the M protein, it exhibits quite complex antigenic diversity. M proteins bind to extracellular matrix (ECM) components such as fibronectin (Fn) to promote GAS colonization. In addition, GAS is internalized into epithelial cells by cell surface integrin α5β1 or CD46, which evades the surveillance of the immune system [15]. M protein can also evoke auto-antibodies, resulting in damage to heart tissue. Moreover, human cardiac myosin has a strong homology with specific fragment sequences in M protein [16]. This means that in individuals with RHD, T cells that are cytotoxic to the myocardium are produced, leading to damage to the target tissue [17]. Lastly, soluble M protein acts as a second signal for caspase-1-dependent NLRP3 inflammasome activation, inducing the maturation and release of proinflammatory cytokine IL-1β, triggering programmed cell death in macrophages [18].

### 2.2. The Streptococcal Cysteine Protease (SpeB)

SpeB has many substrates and can cleave various host and bacterial proteins. For example, SpeB can degrade immunoglobulins IgA, IgM, IgD, IgE, and IgG into small fragments, reducing antibody-mediated regulatory phagocytosis [19]. SpeB can also degrade C3b, a potent opsonin that recruits phagocytes to infected lesions, thereby inhibiting the migration of phagocytes [20]. Furthermore, it has been found that SpeB has two proinflammatory mechanisms: one that directly cuts and activates precursors of IL-1β and IL-36γ [21] and the other that involves entering the infected skin epithelial cells and directly cleaving and activating GSDMA, thereby triggering cellular pyroptosis [22].

### 2.3. Streptococcus C5a Peptidase (SCPA)

SCPA is a serine proteinase with the primary function of inactivating the complement pathway by cleaving C3a and C5a proteins [23]. This effectively impairs the infiltration and activation of neutrophils, which is a key defense mechanism of innate immunity. Meanwhile, SCPA (as a surface binding protein of GAS) can cause diseases such as pharyngitis and tonsillitis [24]. In children with acute pharyngitis, a large number of anti-SCPA antibodies were produced in serum in the later stage of infection, indicating that SCPA is highly immunogenic [25]. In addition, SCPA can fuse with GAC, which can trigger a high degree of T-cell activation.

### 2.4. Streptolysin O (SLO)

Most clinically isolated GAS strains can secrete cytolytic toxins, including SLO and streptolysin S (SLS), whose main function is to cause cell damage by forming pores on the cell membrane. SLO disrupts the host defense mechanisms of macrophages and epithelial cells via Golgi cleavage, promoting GAS intracellular survival and cytotoxicity [26]. In addition to protecting GAS from phagocytosis and killing, SLO has also been shown to promote superantigen penetration of layered squamous cell mucosa, enhance the level of SLO-associated tissue damage during infection, and induce platelet–neutrophil aggregation, leading to vascular occlusion and tissue damage [27].

### 2.5. S. pyogenes Cell-Envelope Proteinase (SpyCEP)

SpyCEP is a conserved and surface-exposed GAS serine protease whose activity is associated with the severity of invasive diseases in humans [28]. In the upper respiratory tract, SpyCEP contributes to the survival of GAS in the nasopharynx but is not necessary [29]. In contrast, the transmission of GAS from the nasopharynx to the lungs is dependent on SpyCEP. SpyCEP can cleave and inactivate all CXC chemokines containing ELR (glutamic acid leucine arginine) motifs, such as CXCL1, CXCL2, CXCL6, etc. More importantly, SpyCEP cleaves CXCL8 (IL-8), providing a mechanism for GAS to escape neutrophil killing [30]. SpyCEP also specifically cleaves antimicrobial peptide LL-37, supporting the survival of GAS infections [31]. Besides its enzymatic activity, SpyCEP can specifically mediate the internalization of GAS into endothelial cells. Thus, SpyCEP is involved in the pathogenesis of GAS infection in multiple ways.

## 3. The History of GAS Vaccines

The first attempts to prevent GAS infection with vaccines by extracting whole cells from heat-killed GAS as immunogens failed to achieve the expected results and resulted in complex inflammatory reactions [32]. Subsequently, Lancefield et al. found that serum antibodies of M protein found in patients with GAS infection persisted for up to 32 years [33]. Therefore, M proteins were considered a potential target as a GAS vaccine candidate. Fox et al. immunized volunteers subcutaneously with purified M protein preparations. Within 2 weeks, 31 out of 33 subjects showed a significant increase in hemagglutination titers with specific types of bactericidal properties. The hemagglutination titer increased by an average of five times compared to the preinjection level [34]. Beachey et al. used highly purified M24 protein extracted via pepsin, which induced specific immunity in both rabbits and guinea pigs [35]. To prevent infection with different serotypes of GAS, the researchers designed N-terminal peptides of M5 and M24 proteins. The peptides were mixed in complete Freund’s adjuvant (CFA) and injected into rabbits, producing specific antibodies against M5 and M24 without cross reaction in heart tissue [36]. Afterward, Beachey et al. designed hybrid peptides with M5, M6, and M24 protein epitopes, namely the trivalent N-terminal M protein vaccine [37]. Dale et al. designed a hexavalent vaccine containing M protein peptides from M24-, M5-, M6-, M19-, M1-, and M3-type GAS and extracted high-titer bactericidal antibodies from experimental animal serum [38]. In the conserved C region of the M protein, Olive et al. fused the J8 peptide segment with the lipid–core–peptide (LCP) system (as adjuvant), and immunized mice were able to induce a strong IgG response, indicating that LCP-J8 has immunogenicity [39]. Later, Batzloff et al. combined the J8 peptide segment with diphtheria toxoid (DT) protein to increase its immunogenicity and found that the vaccine could induce greater IgG responses [40]. In addition to their attempts with respect to M protein, Schulze et al. utilized the H12 fragment of Sfb1, a protein involved in bacterial adhesion, as a mucosal immunogen, which was able to induce high levels of IgG and IgA in mice [41].

Although the above experiments obtained encouraging immunogenicity results, in 1968, Massell et al. used purified M3 protein extract to vaccinate 21 children (all brothers and sisters of ARF patients), which resulted in ARF in two of the vaccinated children, and one case of suspected ARF [42]. Based on this, concern was raised that some of the antigens used in these GAS vaccines may contain antigenic epitopes that trigger an autoimmune response, producing autoantibodies that lead to diseases such as ARF. In 1979, the FDA issued a ban on the use of “GAS components or derivatives” in vaccines, which lasted until 2006, when the ban was lifted, which also led to a stagnation in GAS vaccine development [43].

## 4. Current Status of GAS Vaccines

GAS is a complex pathogen with varying antigenic epitopes and virulence factors among strains [44]. In addition, variation in prevalent strains among regions complicates the targeting of specific GAS proteins to address all infection serotypes. Nevertheless, GAS vaccine development can be broadly divided into two categories: (1) M-protein-based vaccines (in Table 1) and (2) designer vaccines for candidate non-M protein antigens (in Table 2). Here, we focus on GAS vaccine candidates designed around these two aspects in recent years.

### 4.1. M-Protein-Based Vaccines

#### 4.1.1. 26-Valent M-Protein-Based Vaccine

The 26-valent M-protein-based vaccine contains M protein peptide segments (N terminus) from 26 different GAS serotypes. After immunizing rabbits, effective antibodies were produced against 26 different serotypes of GAS [45]. Following this, the antibodies were assessed in 26 healthy adult volunteers. No evidence of ARF was detected, and there was no occurrence of immune sera cross reaction with human tissue [46]. The 26-valent vaccine has since been superseded and replaced with a 30-valent vaccine.

#### 4.1.2. 30-Valent M-Protein-Based Vaccine (StreptAnova)

StreptAnova comprises 4 recombinant proteins and covers 30 GAS serotypes prevalent in North America and Europe (causing common pharyngitis and invasive and/or rheumatic disease) [47]. This vaccine exhibits high immunogenicity in rabbits and can induce antibodies produced by 72 GAS genotypes: 30 genotypes included in the vaccine and 42 genotypes not included [48]. At present, StreptAnova has completed Phase I clinical trials, the results of which showed that the StreptAnova vaccine demonstrated good immunogenicity and tolerability without clinical evidence of autoimmune diseases or laboratory evidence of tissue cross-reactive antibodies. Based on large-scale genomic analysis of GAS vaccine coverage, StreptAnova has a theoretical global coverage of 48%, which may provide coverage for 80.3% of African isolates [49].

#### 4.1.3. StreptInCor

StreptInCor is a 55-amino-acid synthetic polypeptide vaccine with T-cell epitopes (composed of 25-amino-acid residues) and B-cell epitopes (composed of 22-amino-acid residues). Structurally, StreptInCor can bind to different human leukocyte antigen class II molecules, forming a “pocket” of antigen-presenting cells (APCs) [50]. This, in turn, activates T cells via the T-cell receptor (TCR) and stimulates B cells to secrete specific antibodies, which also provides the possibility for StreptInCor to become a universal vaccine. In mice experiments, StreptInCor efficiently induced IgG. StreptInCor-immunized mice showed a high survival rate of up to 87%, while the survival rate of unvaccinated mice was only 53% [51]. In addition to M1 GAS, the StreptInCor vaccine can significantly prevent infection with M5, M12, M22, and M87 GAS strains [52]. The safety of StreptInCor was evaluated in miniature pigs, and no pathological signs of toxicity or abnormalities were observed in the heart and other organs [53]. At present, the vaccine is about to begin Phase I clinical trials.

#### 4.1.4. J8-DT and MJ8Vax

J8 is the smallest epitope in the C region of the M protein, and its binding to diphtheria toxoid (DT, J8-DT) is effective in animal models at preventing infection with various GAS strains [40]. However, the vaccine has almost no effect on the highly virulent CovR/S mutant GAS strain, which can degrade IL-8 and therefore cannot recruit neutrophils [54]. To address this problem, SpyCEP was merged effectively with J8-DT, which is effective in preventing CovR/S mutant GAS strains but has poor immunogenicity [55]. Subsequently, the vaccine was modified by replacing DT with its analog cross-reactive material (CRM, an inactive and non-toxic form of DT). The S2 peptide was modified by lysine residue K4S2 for better dissolution in aqueous solution [56]. Therefore, the J8-CRM with K4S2-CRM peptide conjugate vaccine was called MJ8Vax. In the Phase I clinical trial evaluation of MJ8Vax, it was shown that intramuscular injection of MJ8Vax is safe and effective, but antibody levels decline over time. It was also observed that intramuscular injection of MJ8Vax (adjuvant: alum) did not cause the production of IgA in the respiratory mucosa [57]. To address this issue, the J8 peptide was conjugated to liposomes, significantly reducing the colonization of high-virulence CovR/S mutant GAS in the upper respiratory tract of mice after intranasal infection [58]. At present, the product is still in the preclinical research stage, and if successful, it will be the first nasally administered GAS vaccine.

#### 4.1.5. PMA-P-J8

PMA-P-J8 contains the J8 B-cell epitope of M protein, PADRE (pan HLA-DR-binding epitope) and poly methyl acetate (PMA). PMA-P-J8 induced significant expression of IgG and mucosal IgA after a single immunization in mice, demonstrating strong opsonizing activity against clinical GAS isolates [59]. The advantage of this vaccine is that low doses of oral administration could induce significant systemic and mucosal immune responses without the aid of external adjuvants [59]. At present, the vaccine needs more animal experiments to verify its safety and effectiveness. Overall, this strategy provides a new direction for oral subunit vaccines against GAS.

#### 4.1.6. P*17/K4S2 (CRM) and BP-p*17-S2

P*17 is a derivative of the p145 peptide in the C region of the M protein. Compared with control mice inoculated with p145-DT, a single immunization with P*17-DT significantly enhanced protection against skin and invasive diseases (i.e., >100-fold reduction in skin GAS load and >10,000-fold reduction in blood GAS load) [60]. After being immunized with P*17/K4S2 in combination with CAF^®^01 adjuvant for 10 weeks, mice were infected with highly virulent GAS via nasal or skin injections. The results showed that compared with the control group, the GAS load in the nasal tract (i.e., 85%) and blood (i.e., 94%) was significantly reduced, indicating that P*17/K4S2 (CAF^®^01) can induce effective immunity [61]. Health Canada has approved a Phase I clinical trial of P*17/K4S2 (CRM) [62].

Shuxiong Chen et al. synthesized BP-p*17-S2 using biopolymer particles (BP). In a GAS-infected mouse model, vaccination with BP-p*17-S2 resulted in a significant reduction (>100-fold) in GAS load in nasally associated lymphoid tissue, spleen, and lungs without adverse reactions [63]. The advantages of BP-p*17-S2 are that it can produce vaccines at low cost using endotoxin-free *Escherichia coli* strains as production hosts, it has excellent stability at room temperature, and it causes the induction of antigen-specific humoral and cell-mediated immune responses [63]. Currently, BP-p*17-S2 is in the preclinical research stage.

**Table 1 vaccines-11-01510-t001:** M-protein-based vaccines.

Vaccine Name	Target Antigen	Stage of Development			Adjuvant	Advantage	Ref.
		Preclinical	Phase I	Phase II			
26-valent vaccine	M protein	√	√	√	Alum	Effective against 26 different serotypes of GAS; No occurrence of immune sera cross-reaction with human tissue.	[45,46]
StreptAnova	M protein	√	√		Alum	Effective against 72 GAS emm types; No autoimmune diseases or tissue cross-reactive antibodies.	[48,49]
StreptInCor	M protein	√	√		Alum	Effective against M1, M5, M12, M22, and M87 GAS strains.	[51,52]
MJ8Vax	M protein	√	√		Alum	Significantly induces IgG and IgA; Effective against CovR/S mutant GAS strains.	[40,55]
PMA-P-J8	M protein	√			No	Significantly induces IgG and IgA without adjuvant.	[59]
P*17/K4S2 (CRM)	M protein	√			CAF^®^01	Effective against CovR/S mutant GAS strains.	[60,61]
BP-p*17-S2	M protein	√			Alum	Low cost; Excellent stability at room temperature.	[63]

### 4.2. Non-M-Protein-Based Vaccines

#### 4.2.1. Carbohydrate Vaccines

Group A carbohydrate (GAC) is a cell wall polysaccharide comprising the N-acetylglucosamine (GlcNAc) side chains that makes up ~50% of the cell wall. However, immune cross reactivity between anti-GAC antibodies and host heart valve proteins and cytoskeletal proteins (e.g., actin, keratin, myosin, and vimentin) raises important potential safety issues regarding the use of GAC as a component of GAS vaccines [64,65]. Therefore, after combining the mutant GAC without the GlcNAc (GlcNAc is considered a triggering factor for complications that occur after the GAS infection) side chains with Strep A arginine deiminase (ADI) to form a vaccine, the skin of immunized mice was protected from GAS infection, although not against invasive GAS infections [66]. Nevertheless, due to the conserved nature of GAC, it may become a universal vaccine for GAS. Another new direction involves combining the GAC antigen with different carrier proteins to generate new GAS vaccines. For example, GAC was found to significantly induce specific GAC antibody expression by binding to PADRE [67].

#### 4.2.2. Combo4

Combo4 is a four-component vaccine that includes SLO (a pore-forming toxin), SpyAD (a surface-exposed adhesin), SpyCEP (a protease), and GAC [68]. In the infection models targeting four GAS serotypes (via intranasal or intraperitoneal administration), the survival rate of mice vaccinated with the SLO+SpyAD+SpyCEP/alum vaccine was significantly improved, with the advantage of inducing both bactericidal antibodies and neutralizing hemolysis (SLO and SpyCEP cause cell rupture), as well as inhibiting hydrolysis of IL-8 protein [69]. To improve the coverage of this vaccine, GAC was included as the fourth antigen in the vaccine, namely the Combo4 vaccine. In the mouse infection model, the Combo4 vaccine showed a significant immunoprotective effect and was effective in phagocytic killing assays [68]. At present, the vaccine is in the preclinical research stage.

#### 4.2.3. Combo5

Combo5 is composed of SLO, SpyCEP, Streptococcus C5a peptidase (SCPA), arginine deiminase (ADI), and trigger factor (TF). The Combo5 vaccine can protect mice from skin infections caused by GAS but cannot prevent invasive GAS infections [66]. In a rhesus monkey experiment, the Combo5 vaccine did not prevent throat colonization by GAS infection but effectively alleviated the clinical manifestations of pharyngitis and tonsillitis [70]. When SMQ (containing 3D6AP and QS21) was used as an adjuvant instead of alum, the Combo5 vaccine achieved a survival rate of up to 90% in mice infected with lethal M1T1 strain 5448 GAS. Furthermore, it is important to note that the protection provided by Combo5 is not mediated by opsonic antibodies; instead, Combo5/SMQ immunization leads to the massive secretion of IL-6 and IL-10, as well as Th1-type cytokines (IFN-γ and TNF-α), which causes the Th1 response, balancing Th1/Th2 responses, which is important in preventing invasive GAS infections [71]. At present, the vaccine is in the preclinical research stage.

#### 4.2.4. TeeVax

TeeVax is a recombinant protein vaccine targeting the T antigen of GAS, consisting of TeeVax1, TeeVax2, and TeeVax3, each of which consists of a combination of six specific T-antigen domains [72]. Immunizing rabbits with TeeVax produced IgG that targeted all T-antigen components of GAS. Therefore, the TeeVax vaccine can theoretically achieve a wider coverage of GAS strains. In addition, TeeVax significantly improved the survival rate of mice with lethal invasive GAS infection [72]. Currently, TeeVax is in the preclinical testing stage, using different adjuvants to elicit humoral and cellular immunity.

#### 4.2.5. VAX-A1

VAX-A1 was recombined from GAC^PR^–SpyAD conjugates with SLO and SPCA proteins. GAC^PR^ is a form of GAC that lacks GlcNAc side chains [70]. The antiserum produced by immunizing rabbits with VAX-A1 promotes phagocytic clearance of multiple GAS serotypes in vitro. In both systemic and local skin infection models, VAX-A1 protected mice from attacks by GAS. Furthermore, in vitro Western blot analysis revealed no additional cross reactivity between antiserum and lysates from the human heart or brain [73]. An application for “Investigational New Drug” was submitted for VAX-A1 in 2022.

#### 4.2.6. 5CP

5CP is a five-protein recombination vaccine consisting of SLO, SpyAD, SpyCEP, SCPA, and sortase A. In an investigation of serum samples from 62 children, each type of antigen in 5CP was found to be able to induce an effective IgG response, indicating that 5CP may exhibit antigenicity in humans [74]. Immunized mice were infected with lethal doses of M1 or M49 GAS strains, and the survival rate of 5CP-immunized mice was as high as 95% [74]. Intranasal immunization of mice with 5CP and CpG-oligodeoxynucleotide (i.e., CpG as adjuvant) effectively protected against mucosal and systemic infection by different serotypes of GAS, lasting for at least 6 months [71]. The 5CP/CpG combination inhibited the development of subcutaneous lesions, promoted lesion recovery, and protected subcutaneous invasive disease models. In addition, 5CP-immunized mice induced Th17 cell response (Th17 cells have an important role in anti-GAS immunity and contain T-cell epitopes in humans) [75]. The vaccine has not yet been systematically evaluated.

#### 4.2.7. Spy7

Spy7 is recombined from seven proteins: SpyAD, PulA, OppA, SCPA, Spy0843, Spy1037, and Spy1228. The rationale for the Spy7 vaccine design stems from the fact that intravenous immunoglobulin G (IVIG) provided effective protection during invasive GAS infections. The Spy7 vaccine induced anti-GAS antibody production and T-cell response in mice and limited the transmission of M1 and M3 GAS from infected lesions, reducing the severity of disease caused by M1 GAS (but not M3 GAS) [76]. The vaccine has not yet been systematically evaluated.

#### 4.2.8. SPy_2191

Reverse vaccinology is the use of bioinformatics technology to analyze microbial genome sequences and to screen candidate antigens for vaccine development from a large number of antigens without the need for pathogen cultivation. Based on the reverse-vaccinology method, Pooja Sanduja et al. identified SPy_2191 as a cross-protection vaccine candidate [77]. SPy_2191 is conserved in GAS, is surface-exposed, and inhibits GAS adhesion and invasion. Mice immunized with SPy_2191 were able to induce large amounts of bactericidal antibodies that effectively killed strains obtained from multiple countries (M1 and M49 GAS in India, DM3.1 in Israel, and M1 GAS in the United States and the United Kingdom). When mice were infected with invasive GAS serotypes, SPy_ 2191 improved the survival rate by up to 92%, significantly reducing the burden of GAS in organs. In addition, SPy_2191 significantly inhibited the pharyngeal colonization of GAS in a mouse mucosal infection model. In summary, SPy_2191 can be used as a universal vaccine candidate against GAS infection [77]. However, more animal studies are needed to test the safety and effectiveness of SPy_2191.

**Table 2 vaccines-11-01510-t002:** Non-M-protein-based vaccine.

Vaccine Name	Target Antigen	Stage of Development			Adjuvant	Advantage	Ref.
		Preclinical	Phase I	Phase II			
GAC	GAC without GlcNAc side chain	√			CFA	Immunized mice were protected from GAS infection, although not against invasive GAS infections.	[66]
Combo4	SpyCEP, SLO, SpyAD, GAC	√			Alum	Induction of both bactericidal antibodies and neutralizing hemolysis, as well as inhibition of hydrolysis of IL-8 protein.	[69]
Combo5	SLO, SpyCEP, SCPA, ADI, TF	√			SMQ	Effective against lethal M1T1^5448^ GAS; Combo 5 caused a Th1 response.	[70,71]
TeeVax	T antigen	√			Alum	TeeVax targeted all T-antigen components of GAS.	[72]
VAX-A1	GAC^PR^, SpyAD, SLO, SPCA	√			Alum	No additional cross-reactivity between antiserum and lysates from the human heart or brain.	[73]
5CP	SrtA, SCPA, SpyAD, SpyCEP, SLO	√			CpG	5CP induced Th17 responses.	[74,75]
Spy7	SCPA, OppA, PulA, SpyAD, Spy1228, Spy1037, Spy0843	√			Alum	Spy7 induced T-cell responses.	[76]
SPy_2191	SPy_2191	√			Alum	SPy_2191 induced bactericidal antibodies that effectively killed strains endemic in multiple countries.	[77]

ADI: arginine deiminase; SCPA: Streptococcus C5a peptidase; TF: trigger factor; SMQ: squalene-in-water emulsion containing a Toll-like receptor 4 agonist and QS21; GAC: group A carbohydrate; SLO: streptolysin O; CpG: CpG-oligodeoxynucleotide; CFA: complete Freund’s adjuvant.

## 5. Challenges and Prospects for GAS Vaccine Research and Development

In 2018, the World Health Assembly adopted an important resolution on better control and prevention of GAS infection and extracted a development strategy for prioritization of the development of GAS vaccines [12]. Therefore, the prospects for GAS vaccine development are more promising today than before 2018, with several potential vaccine candidates already making substantial progress in clinical trials. However, there are still many obstacles to the development of GAS vaccines, including the following [62,78,79]: (1) a lack of in-depth understanding of the intricate molecular mechanisms associated with complications caused by GAS infection, (2) a paucity of suitable animal models susceptible to GAS infection for vaccine assessment, (3) a lack of a strategy to improve vaccine coverage for multiple serotypes, (4) insufficient advocacy work on GAS-induced deaths, (5) high research and development costs for innovative vaccines, (6) a lack of epidemiological and economic statistical data from underdeveloped countries, (7) the absence of universally standardized assessment criteria, (8) the complexity of GAS epidemiology and the potential problem of drug resistance, and (9) the issue of how to avoid the risk of autoimmune complications from vaccines.

As for the vaccine itself, it consists of immunogens, adjuvants, and carriers. The immunogens determine the specificity and targeting of the induced immune response, adjuvants determine the intensity of the immune response, and the carriers determine the type of immune response. At present, the main adjuvant for GAS vaccines is alum. Recent evidence suggests that the selection of adjuvant type is crucial for the immune response induced by GAS vaccines [80]. Optimization of the combination of antigens and adjuvants is an important component in the development of GAS vaccines. Vaccine carrier systems can modify the immune response of traditional vaccines and optimize the effectiveness of vaccination [81,82]. Thus, developing new carrier systems such as liposomes, microneedles, viral vectors, and extracellular vesicles could further accelerate the success of GAS vaccines. For example, microarray patch vaccines delivered through the skin have the advantages of potential dose saving and ease of use compared to traditional intramuscular vaccines [83]. In this regard, the J8-DT candidate vaccine has recently been evaluated for efficacy using high-density microarray patches [84].

Genetic engineering of vaccines refers to the use of gene recombination technology to modify the genome of pathogenic microorganisms in order to reduce their pathogenicity and enhance their immunogenicity. Drawing insights from the success of SARS-CoV-2 vaccine development, the focus is on creating mRNA vaccines tailored to GAS-related infections [85,86]. These mRNA vaccines exhibit immune response mechanisms akin to live pathogens without causing infection and offer enhanced stability, along with efficient antigen protein expression, setting them apart from conventional vaccines. Moreover, nanoparticle vaccines, referring to vaccines that use nanomaterials as carriers, are being developed that connect specific antigens and adjuvants through physical or chemical methods for disease treatment and prevention [87]. The nanocarrier-based delivery system not only protects the vaccine from premature degradation and enhances its stability but also facilitates the targeted delivery of immunogens to antigen-presenting cells [88]. In addition, promoting the integration of multiple disciplines (such as structural biology, reverse genetics, materials science, and artificial intelligence) allows us to better understand how vaccines stimulate the body’s immune response, select target molecules, and better predict possible adverse reactions. This will lead to a more rational, accurate, and efficient design of GAS vaccines.

## 6. Conclusions

GAS infection is becoming an increasingly common health problem today, especially in economically disadvantaged countries and regions. Therefore, there is an urgent need to accelerate the development of safe and effective GAS vaccines. At present, the WHO and other organizations are vigorously supporting the development and implementation of GAS vaccines, encouraging researchers to collaborate across disciplines in the search for richer and more effective epitopes [62,89]. Given the challenges of GAS vaccine development outlined in this review, including the widespread genetic heterogeneity and highly variable protein sequences, there is a need to strengthen basic research into GAS vaccines. This will improve the ability to develop more effective and targeted vaccine strategies for this pathogen.

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
