# Peer review of "Streptococcus pyogenes: Pathogenesis and the Current Status of Vaccines"

_vaccines, 2023, doi:10.3390/vaccines11091510_

Round 1

Reviewer 1 Report

This manuscript is an useful contribution to the important research topic of vaccine development targeting Streptococcus pyogenes. However, some corrections and improvements are necessary as detailed in the following and in the enclosed Pdf-file:

1. Although many scientists prefer to use the term "group A streptococcus" for Streptococcus pyogenes (the exact taxonomic name) one should prefer S. pyogenes in the scientific literature, since there are strains of S. dysgalactiae subsp. equisimilis carrying the identical group A cell wall polysaccharide antigen like S. pyogenes. Therefore I propose to change the title of the manuscript as followows:

"Streptococcus pyogenes: Pathogenesis and the Current Status of Vaccines"

2. The list of References must be revised regarding the following aspects:

a) In several references the names of the authors (with respect to first name and surname) are not correctly cited, e. g. ref. no. 42 This has to be corrected according to the rules of the journal and informations found in international databases and the full texts of the publications.

b) There are two references listed in duplicate: Refs. nos. 54 and 58, 3 and 69.

c) In some references a "[J]" is found before the name of the journal. This should be corrected. In ref. no. 52 (Burns et al.) no journal name is given; "NPJ Vaccines" should be added.

Further proposals for changes and corrections can be found in Pdf-file enclosed.

There is room for improvements in the English language. I have made some proposals in the enclosed Pdf-file.

Reviewer 2 Report

The authors are to be commended for their effort to comprehensively review group A Strep pathogenesis and vaccine development which can be a daunting task even for experts in the field.  Please see suggestions below:

1) Abstract (and throughout paper): Streptococcus should be capitalized and italicized when used in "group A Streptococcus

2) Abstract, line 12.  Manifestation should be plural - "manifestations"

3) Abstract, lines 14-16.  The intended meaning of the sentence beginning "Although the development of antibiotics.." is not clear, nor is it apparent that this is a critical point that needs to be emphasized given the focus of the paper.  Suggest rewording or deleting this sentence.

4) Abstract, line 16:  "..affected the efficacy of traditional antibiotics" is not accurate.  Penicillin is the traditional antibiotic of choice for GAS and has not yet been affected by drug resistance, although this needs to be closely monitored for.  It is important second-line agents such as Macrolides and Clindamycin for which resistance is an issue.

5) Abstract, line 17.  Suggest replacing "research" with "surveillance in this line.

6) Abstract, line 19 (and throughout paper), the "which" after the comma here is unnecessary and can be deleted.  Sugggest looking throughout paper for where "which" is used and assessing if it is needed.

7) Abstract, lines 29-32.  The language "that we believe are worth investigating" is subjective and non-informative for reader.  If possible, please provide more objective rationale and language for why certain vaccines are the focus (e.g. they are further along in development, showed most promise in pre-clinical studies etc.)

8)Introduction, line 38.  "coccus" here is not a specific scientific notation and does not need to be italicized

9) Introduction, line 39.  Suggest "superficial" rather than purulent.

10) Intro, line 40.  "Toxin-mediated" rather than "Toxic"

11) Intro, line 41, "Immune-mediated" rather than "allergic"

12) Intro, lines 39-41.  Missing several large categories of GAS diseases such as invasive infections and superficial skin infections

13) Intro lines 46-48.  The sentence "The main causes of human death caused by GAS are autoimmune sequelae (e.g., ARF, RHD) and antibiotic resistance caused by repeated infections of GAS[2,4]" is not accurate.  Apart from RHD, the other main cause of death from GAS is invasive disease not necessarily caused by antibiotic resistant organisms (see paper by Carapetis et al, Lancet ID 2005). Also not demonstrated that repeated infections causes resistance.

14) Intro, lines 62-63.  Delete scarlet fever in parenthesis.  Scarlet fever is not an invasive GAS infection

15) Intro, lines 63-65.  Not accurate to say the increase in invasive GAS "mainly" affected children.  It disproportionately affected children, but most cases were still in adults.

16) Lines 65-67.  For number of deaths in US, 1) this is an annual estimate not a total between 2005-2012; 2) more recent annual estimates are available at https://www.cdc.gov/abcs/downloads/GAS_Surveillance_Report_2019.pdf

17) Line 72-74.  "However, the continuous occurrence of treatment failure cases poses a threat to the reliance on penicillin-based antibiotics to treat GAS infections."  This statement is not based on data and should be removed.

18) Line 76, replace "injection" with "vaccination"

19) Line 84-85.  delete phrase "with a special focus on several potential candidate vaccines" or provide a clear rationale for emphasis on specific vaccines.

20) Line 91, cause should be plural.

21) line 95.  "which is" is unnecessary

22) lines 104-105.  M-protein does not produce auto-antibodies, it elicits their production

23) lines 104-106, Statement "The M protein can also produce auto-antibodies, leading to dysfunction of normal heart tissue" requires a supportive reference.  Typically understood that these auto-antibodies result in damage to heart tissue (resulting in abnormal tissue) rather than dysfunction.  

24) lines 106-107,  Although this is understood to be true, suggest providing reference for this sentence "Moreover, human cardiac myosin has strong homology with specific fragment sequences in the M protein."

25) lines 119-121.  Sentence starting "SpeB can also degrade C3b.." should have reference.

26) lines 126-131, summary of functions of SCPA very brief and does not encapsulate all of its known functions.  Suggest expanding this section; also more commonly referred to as a proteinase rather than peptidase

27) lines 146-148, first sentence of SpyCEP sub-section should have a reference provided.

28) lines 148-150, The first part of this sentence that SpyCEP can cause necrotizing infections needs a reference.  The second part of this sentence that this is the basis of respiratory transmission does not make sense.  Respiratory transmission is well documented to occur in the absence of necrotizing soft-tissue infections.

29) lines 158.  "is" not "was involved

30) line 161-164 (first sentence in history of vaccine section) would benefit from reference

31) line 166, redundant "which"

32) lines 170-171 would benefit from additional specificity on which particular antibodies increased and by how much

33) lines 171-173.  Beachey reference provided for this statement only includes data from rabbits and guinea pigs, not humans.  Reword sentence to reflect this or add additional reference that provides human data

34) line 190 - "toxoid" rather than "toxoido"

35) lines 205-206.  Sentence " However, the generation of antibodies was exactly what the 205 vaccines were originally designed to do" should be deleted as not relevant since vaccines are not designed to generate auto-antibodies which was the concern.

36) lines 213-215, reference should be provided for this sentence on strain variation.  Many suitable references exist, for example Steer AC, Law I, Matatolu L, Beall BW, Carapetis JR. Global emm type distribution of group A streptococci: systematic review and implications for vaccine development. Lancet Infect Dis 2009; 9:611–6.

37) Line 225-226, reference 36 should follow this line

38) Line 226-227, reference 37 should follow this line

39) lines 229-230, this sentence is inaccurate and should be deleted.  The 26-valent vaccine has been superseded and replaced by the 30 valent vaccine.  Phase II trials of the 26 vaccine are not underway currently.

40) lines 235-238, reference needed for these lines

41) lines 238-242.  Reference needed

42) line 243, Replace StreptAnova's with StreptAnova

43) lines 242-245.  Please check references here, correct reference for 80.3% number appears to be paper from Taariq Salie and colleagues in mSphere "Systematic Review and Meta-analysis of the Prevalence of Group A Streptococcal emm Clusters in Africa To Inform Vaccine Development"

44) lines 256-258, reference needed for results of mice experiments

45) lines 265-7, reference(s) needed

46) lines 276-8.  reference needed

47) Line 221, not all the M-protein based vaccines are recombinant.  Better to refer to them as M-protein based and note in the individual descriptions which are recombinant

48) lines 295-8, reference needed

49) line 313-317.  Unclear how the reduction in GAS load in nasal tract and blood lead to conclusion for potential for long-term immunity.  Rationale for this statement needs to be clarified

50) line 317-319, reference needed, also not aware of a group called "Canadian Department of Health" - please confirm this name

51) line 321.  Please specify which animal in this model

52) lines 321-5, reference needed

53) line 324.  The word "characteristic" here seems out of place since all vaccines have multiple characteristics.  Suggest phrasing such as "Advantages of BP-p*17-S2 are ..."

54) Table 1, the against 72 GAS serotypes of StreptAnova should indicate that it has been shown to induce antibodies against these emm types.  Also genotype and serotype used variably, suggest using emm types throughout

55) Line 347.  Unclear what is meant by "conservative" in this sentence, suggest switching "conservative and universality" to "conserved nature"

56) line 399.  "detect" is not right word to describe role of adjuvants, "elicit"?

57) line 332.  Delete "recombinant" from the title and change vaccine to plural

58) lines 413-416, reference needed

59) lines 418-425, reference needed

60) lines 426-427.  Unclear why a rapid resolution of Th17 response is necessary, if keep the statement in parentheses suggest briefly explaining why rapid resolution is required.

61) line 428-9, is there a reference for this statement?

62) lines 448-450.  Suggest rewording to "effectively killed strains obtained from multiple countries (M1..."

63) Table 2, need to define in footnotes to table the abbreviations used in the table 

64) lines 467-470.  Unclear why year 2016 is specifically mentioned given WHO resolution was in 2018

65) Lines 470-484.  This list of obstacles should follow a colon and the obtacles should be separated by a semi-colon not a period

66) Line 474, given that multiple animal models have been referenced in the paper, seems odd to refer now to an "absence of suitable animal models".  Suggest referring instead to a "paucity" or "Shortage"

67) Lines 475-476, unclear what is meant by "the issue of GAS vaccine coverage", needs to be reworded for clarity of meaning

68) lines 476-477, "publicity is not the right word here.  Suggest "advocacy".  Also "injuries" is not the right word

69) lines 477-479, The intended meaning of the phrase "The disparity between investment in vaccine research and development versus the commercial return rate" is not clear.  Please reword to make intended meaning more clear

70) lines 482-483.  Meaning of "potential problem of variability" is uncertain.  Variability of what?  Needs to be clarified

71) line 488, adjuvants are one contributing determinant to the intensity of the immune response, not the only one

72) lines 497-500.  Replace "will" in this sentence with "could", as there is no certainty that new carrier systems will guarantee the success of GAS vaccine

73) lines 500-502, reference needed

74) line 506-507.  Suggest rewording this sentence, hard to follow meaning as currently written

75) line 527.  Suggest replacing "reasonable" with "rational"

76) line 534.  Suggest deleting word "vaccine" from in front of organizations

77) line 540-541.  "there is a call to.." is confusing as worded.  Call from whom?  Better to replace "call" with "need"?

78) References, formatting is not consistent throughout with some having full names of authors and others not.  Please ensure that is the same for all references in agreement with journal style

While in general the quality of English language throughout the paper is fairly good, there are common mistakes that are repeated that the authors should be careful to carefully check for in any revision.  These include failure to italicize Streptococcus in group A Streptococcus and the unnecessary use of the word "which" in many sentences.  I have commented on these early in the paper but not on every occasion when they recur. 

Round 2

Reviewer 2 Report

Thank you for the diligent efforts to respond to all reviewer and editor comments and improving the manuscript.  Some additional suggestions below:

1)  Suggest modifying line 14-16 in Abstract to: "At present, although GAS is still sensitive to penicillin, there are cases of treatment failure for GAS pharyngitis and antibiotic therapy does not universally prevent subsequent disease."

2) Suggest deleting "showed most promise in pre-clinical studies" in line 31 of abstract

3) Line 42-43, Add bacteremia and meningitis as other examples of invasive infections

4) Line 83-84, change to "development, and have showed the most promise in pre-clinical studies"

5) Lines 126-136,  Please  provide references for lines 132-134, and lines 134-136.

6) Lines 152-154 needs supportive reference

7) Lines 154-155 needs supportive reference

8) Lines 175-177, "but would benefit from additional specificity on which particular antibodies increased and by how much" was a comment to authors not a suggestion to incorporate this wording into the text.  Question is which type of antibodies increased and by how much (if data available).  From the Fox reference, it looks like they are referring to type specific hemagglutinin titers.  Please revise this sentence to include any additional available detail on this increase in antibody titers.

9) Line 204, please insert "vaccinate" in front of "21 children"

10) Line 210, suggest deleting "of the irreconcilable conflict between the two"

11) Line 233.  Replace "the" with "a" in front of 30-valent vaccine

12) Line 310, delete "The" in front of "Health Canada"

13) Line 478, suggest add "for multiple serotypes" after "improve vaccine coverage"

14) Line 480, delete "authentic"

15) Line 542-544, suggest rewording to "This will improve the ability to develop more effective and targeted vaccine strategies for this pathogen."

Quality of English language has been improved throughout manuscript.

Author Response

15/9/2023

RE: Manuscript ID: vaccines-2586492

Dear Reviewer,  

We sincerely thank the reviewers for their suggestions on our work, which will effectively improve our review work.

The following is a point-by-point reply that specifically addresses the concerns raised by the reviewer #2. All changes are highlighted in the marked copy of the revised manuscript. Thanks!

Reviewer #2:(Note: The revisions were highlighted by yellow.  )

1.Suggest modifying line 14-16 in Abstract to: "At present, although GAS is still sensitive to penicillin, there are cases of treatment failure for GAS pharyngitis and antibiotic therapy does not universally prevent subsequent disease."
Response: Thank you. We have changed it according to your opinion.(line 14-16)

2.Suggest deleting "showed most promise in pre-clinical studies" in line 31 of abstract.

Response: Thank you. We have changed it according to your opinion.(line 31)

3.Line 42-43, Add bacteremia and meningitis as other examples of invasive infections.

Response: Thank you. We have rewritten it according to your opinion.(line 42-43)

4.Line 83-84, change to "development, and have showed the most promise in pre-clinical studies"

Response: Thank you. We have changed it according to your opinion.(line 83-84)

5.Lines 126-136, Please provide references for lines 132-134, and lines 134-136.

Response: Thank you. We have updated the reference#24 and 25. (line 134 and 137)

6.Lines 152-154 needs supportive reference.

Response: Thank you. We have updated the reference#28. (line 154)

7.Lines 154-155 needs supportive reference.

Response: Thank you. We have updated the reference#29. (line 156)

8.Lines 175-177, "but would benefit from additional specificity on which particular antibodies increased and by how much" was a comment to authors not a suggestion to incorporate this wording into the text. Question is which type of antibodies increased and by how much (if data available). From the Fox reference, it looks like they are referring to type specific hemagglutinin titers. Please revise this sentence to include any additional available detail on this increase in antibody titers.

Response: Thank you. We have rewritten it according to your opinion. (line 176-181)

9.Line 204, please insert "vaccinate" in front of "21 children"

Response: Thank you. We have changed it according to your opinion.(line 209)

10.Line 210, suggest deleting "of the irreconcilable conflict between the two"

Response: Thank you. We have changed it according to your opinion.(line 215)

11.Line 233. Replace "the" with "a" in front of 30-valent vaccine.

Response: Thank you. We have changed it according to your opinion.(line 238)

12.Line 310, delete "The" in front of "Health Canada"

Response: Thank you. We have changed it according to your opinion.(line 325)

13.Line 478, suggest add "for multiple serotypes" after "improve vaccine coverage"

Response: Thank you. We have changed it according to your opinion.(line 486-487)

14.Line 480, delete "authentic"

Response: Thank you. We have changed it according to your opinion.(line 489)

15.Line 542-544, suggest rewording to "This will improve the ability to develop more effective and targeted vaccine strategies for this pathogen."

Response: Thank you. We have changed it according to your opinion.(line 552-554)

We sincerely hope that these changes are clear and satisfactory enough to permit the acceptance of this manuscript for publication by Vaccines.

Thank you very much for your consideration.      

Sincerely,

Corresponding author: Lin Wei, Email: [email protected]
